# A realist review to explore how low-income pregnant women use food vouchers from the UK's Healthy Start programme

Heather Ohly, Nicola Crossland, Fiona Dykes, Nicola Lowe, Victoria Hall-Moran

► Prepublication history and additional material are available. To view these files please visit the journal online (http://dx.doi.org/ 10.1136/ bmjopen-2016-013731).

## ABSTRACT

**Objectives** To explore how low-income pregnant women use Healthy Start food vouchers, the potential impacts of the programme, and which women might experience these impacts and why.

**Design** A realist review.

**Eligibility criteria for selecting studies** Primary or empirical studies (of any design) were included if they contributed relevant evidence or insights about how low-income women use food vouchers from the Healthy Start (UK) or the Special Supplemental Nutrition Program for Women, Infants and Children (WIC) programmes. The assessment of 'relevance' was deliberately broad to ensure that reviewers remained open to new ideas from a variety of sources of evidence.

**Analysis** A combination of evidence synthesis and realist analysis techniques was used to modify, refine and substantiate programme theories, which were constructed as explanatory 'context–mechanism–outcome'– configurations.

**Results** 38 primary studies were included in this review: four studies on Healthy Start and 34 studies on WIC. Two main outcome strands were identified: dietary improvements (intended) and financial assistance (unintended). Three evidence-informed programme theories were proposed to explain how aspects of context (and mechanisms) may generate these outcomes: the 'relative value' of healthy eating (prioritisation of resources); retailer discretion (pressure to 'bend the rules'); the influence of other family members (disempowerment).

**Conclusions** This realist review suggests that some low-income pregnant women may use Healthy Start vouchers to increase their consumption of fruits and vegetables and plain cow's milk, whereas others may use them to reduce food expenditure and save money for other things.

## BACKGROUND

Healthy Start is the UK government's food voucher programme for low-income pregnant women and young children. It was introduced in 2006, after the Acheson Review drew attention to income as one of the major determinants of health (and nutrition) inequalities, and highlighted the importance of nutrition for women of childbearing

age and their children.[1] The Committee on Medical Aspects of Food and Nutrition Policy was asked by the government to review the long-standing Welfare Food Scheme, which was subsequently replaced by Healthy Start.[2]

Women are eligible for Healthy Start if they receive income-related benefits or child tax credit and an annual household income of £16 190 or less. Pregnant women aged under 18 are eligible regardless of their income. The weekly voucher values are: one voucher per week during pregnancy (£3.10); two vouchers per week for each baby under 1 year (£6.20) and one voucher per week for each child aged 1–4 years (£3.10). The vouchers can be exchanged for fruits and vegetables, plain cow's milk or infant formula. Retailers must be registered with the scheme to accept and claim payment for the vouchers. Healthy Start also offers free vitamins for eligible women and children, but there have been problems with uptake of vitamins.[3] Some areas offer free vitamins to all pregnant women and young children and the option of universalising Healthy Start vitamins remains under review (at the time of writing) by the chief medical officer. Therefore, this review focused on the food voucher component of the programme and low-income pregnant women as the first beneficiaries.

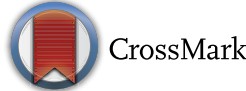

College of Health and Wellbeing, University of Central Lancashire, Preston, UK

**Correspondence to**
Heather Ohly;
HOhly@uclan.ac.uk

Healthy Start aims to provide a 'basic nutritional safety net' and to encourage 'women and families to make positive nutritional choices affecting their longer term health' (p 4).[4] Therein lies an implicit programme assumption that women will use the vouchers to purchase target foods (ideally in greater quantities than they did before) leading to dietary improvements. The vouchers may have been conceived as a financial incentive for dietary improvements, although this was not explicit in policy documentation. Since the introduction of Healthy Start in 2006, there has been no robust evaluation of its impact on nutritional outcomes—despite recommendations published in an early scoping review.[5] However, qualitative studies have indicated a range of experiences, motivations and perceived outcomes, with not all low-income women using the vouchers to improve their diets.[6 7] This review was undertaken to explore how low-income pregnant women use Healthy Start food vouchers, the potential impacts of the programme, which women might experience these impacts and why.

The realist approach was adopted because it is based on the understanding that different individuals or groups of individuals are likely to respond to any programme (or intervention) in different ways. It is a theory-driven approach that seeks to explore outcome patterns (or demiregularities) and offer plausible explanations for how and why they occur. The purpose of realist synthesis is to 'articulate underlying programme theories and then to interrogate the existing evidence to find out whether and where these theories are pertinent and productive' (p 74).[8] The stages of conducting a realist review tend to be iterative and overlapping—a gradual process of developing, testing and refining programme theories. Evidence may be obtained from studies of the programme itself, or more widely from similar programmes that are thought to work in similar ways. Reviewers may adapt and modify existing theoretical frameworks or 'middle-range' theories to help develop their own explanations. The unit of analysis is the programme theory (rather than the specific programme) and this can be considered at different levels of abstraction.

Programme theories are often constructed as 'context–mechanism–outcome' (CMO) configurations, and evidence is sought to substantiate the causal linkages. The logic of realist explanation is that outcomes are caused by mechanisms, and mechanisms may (or may not) be 'triggered' in certain contexts.[8] Context refers to the pre-existing conditions into which the programme is introduced, and there are four layers of context: individual, interpersonal, institutional and infrastructural. Mechanisms are defined as the reasoning and reactions of individuals in response to the resources offered by the programme. A core principle of realism is that mechanisms generate outcomes—they are not a direct result of the programme.

This realist review aimed to explore the following questions:

1. How do low-income pregnant women use Healthy Start vouchers?

2. What are the intended and unintended outcomes of the programme?

3. What are the underlying mechanisms and how do variations in context influence (enable or constrain) these mechanisms?

A preliminary search confirmed the paucity of empirical studies on Healthy Start and we felt that additional sources of empirical evidence would be needed to explore these research questions. The most obvious source of potentially relevant evidence was the Special Supplemental Nutrition Program for Women, Infants and Children (WIC) in the USA. WIC is the only other national food subsidy programme for low-income women of childbearing age, including pregnant and postpartum women and young children. It was first introduced in 1972 and revised in 2009 to reflect current dietary guidelines.[9] The WIC food package for pregnant women is different to Healthy Start, with 'maximum monthly allowances' for a range of foods (reduced-fat milk, whole grain cereals, eggs and pulses) and 'cash value vouchers' for fruits and vegetables. The other main programme difference is mandatory WIC nutrition education for all beneficiaries. There are also many contextual differences between the USA and the UK, such as sociodemographic, cultural, geographical and political characteristics. Despite these differences, there are also likely to be similarities in terms of how low-income women respond to the programme (ie, mechanisms).

## METHODS

### Study protocol and ethical approval

The protocol for this realist review was registered with the International Prospective Register of Systematic Reviews (PROSPERO 2014: CRD42014015050). There were no changes to the review process since this protocol was published. Ethical approval was obtained from the University of Central Lancashire Science, Technology, Engineering, Medicine and Health Ethics Committee in May 2015.

### Programme theory development

This review is about a programme that already exists, with implicit theories and assumptions about how it works and what effects it may have on beneficiaries.[8] Therefore, we used a 'bottom up' approach to theory development in this review, as described by Shearn and Allmark (Realist Research Seminar Series, Sheffield Hallam University, 2016). In other words, we developed theories using information about the Healthy Start food voucher programme rather than theorising at a more abstract level. Candidate theories (or initial, untested theories) were identified and prioritised using information derived from academic and grey literature on Healthy Start, an intervention mapping exercise, existing knowledge, creative thinking, consultations with stakeholders (in person and by email) and discussions among the review team. The stakeholder group included six midwives, two academics and two

public health practitioners, who all shared first-hand experiences and insights about how the Healthy Start programme works in practice.

We developed candidate theories about how low-income pregnant women might experience Healthy Start and what contextual factors might influence their reasoning and reactions (mechanisms) at each of the following stages: access to the programme starting at the first trimester antenatal appointment (eg, issues around clear communication and understanding of eligibility criteria and entitlement); the application process (eg, barriers and facilitators to successfully receiving the vouchers); whether and where women use the food vouchers (eg, issues around convenience and stigmatisation); and how women use the food vouchers (eg, to buy more of the target foods or to save money). We decided to prioritise the last stage of the programme pathway: once an eligible pregnant woman has received the food vouchers, how does she use them and why? This decision reflected the research priorities identified from the literature, the interests of the research team and the time and resources available to conduct this review. The candidate theories we tested were proposed explanations for why women might use Healthy Start to improve their diets during pregnancy, such as motivations and values relating to health benefits. We also considered reasons why women might use their vouchers in alternative ways, including prioritisation of resources, pressure to bend the rules and disempowerment.

### Search strategy

Separate searches were conducted for Healthy Start and WIC:

#### Healthy Start

Studies were identified through manual, purposive, snowball and citation searches (January to March 2015). The search terms used initially were 'Healthy Start' and 'UK' because there is another programme called Healthy Start in the USA, which aims to prevent infant mortality. More precise search terms were not needed due to the paucity of empirical studies and familiarity with the literature.

#### Program for Women, Infants and Children

A broad search strategy was devised in collaboration with an information specialist in the Collaboration for Leadership in Applied Health Research and Care North West Coast. This strategy was adapted and run in six electronic databases in September 2015: MEDLINE, EMBASE, CINAHL, Open Grey, ETHOS and PubMed. Table 1 shows the search terms used in MEDLINE. No date or language restrictions were used. Reference lists of included studies were checked for additional studies. An online list of WIC studies was also checked for additional studies.[10]

#### Inclusion criteria

Primary or empirical studies (of any design) were included if they contributed relevant evidence or insights about how low-income women use food vouchers from

**Table 1** Search strategy used in MEDLINE to identify women, infants and children (WIC) studies

| # | Search terms | Results |
|---|---|---|
| 1 | WIC.tw. | 1008 |
| 2 | (nutrition or food or voucher or 'nutrition program').tw. | 377 002 |
| 3 | 1 and 2 | 599 |
| 4 | (Special* adj4 Supplement* adj4 Nutrit* adj4 Program* adj4 Women* adj4 Infant* adj4 Child*).tw. | 415 |
| 5 | 3 or 4 | 688 |

the Healthy Start (UK) or WIC (USA) programmes. An assessment of 'relevance' is essential in realist synthesis to ensure that all included studies contribute to theory development, refinement and testing.[8 11] In this review, the interpretation of 'relevance' was deliberately broad to ensure that reviewers remained open to new ideas from a variety of sources of evidence. A bespoke system was used to maintain a consistent and transparent approach. Table 2 shows the questions used to assess relevance. These questions were developed by the review team and finalised towards the end of the theory development stage to ensure they reflected the candidate theories we wanted to test. Studies that scored 5/8 or more (based on the total number of yes answers) were included.

### Study selection

Results from the WIC database searches were uploaded into RefWorks (web version; ProQuest; Michigan, USA) and screened using titles and abstracts. Studies that appeared to meet the inclusion criteria were obtained

**Table 2** Questions used to assess the relevance of primary studies

| # | Question |
|---|---|
| 1 | Do the research questions or study aims refer to Healthy Start or Women, Infants and Children (WIC)? |
| 2 | Does the study focus on the food voucher (cash value voucher or food package for WIC) component of the programme? |
| 3 | Does the study focus on beneficiaries (women who were receiving the vouchers) rather than eligibility status (women who were eligible to receive the vouchers)? |
| 4 | Does the sample include pregnant women? |
| 5 | If the sample does not include pregnant women, could some of the findings be generalisable to pregnant women? |
| 6 | Does the study report women's food or nutrient intakes (measured or perceived)? |
| 7 | Does the study provide any insights about how food vouchers are used? |
| 8 | Does the study provide any insights about which women may benefit most/least and why? |

as full-text articles. Studies for which insufficient information was available to determine relevance were also obtained as full-text articles. The full-text screening process was fully documented using Microsoft Excel 2013 V.15.0.4815.1001, including assessments of relevance and reasons for exclusions. The same criteria were applied to studies of Healthy Start. Study selection was completed by the lead reviewer (HO) and double checked by a second reviewer (VM). Any disagreements were resolved by discussion.

### Data extraction
Quantitative data on women's nutritional outcomes were extracted using bespoke tables in Microsoft Word 2013 V.15.0.4815.1001. Other non-relevant data were not extracted. Qualitative data, textual descriptions of findings and author interpretations were extracted using MAXQDA V.11. A coding system was created with three main headings: context, mechanisms and outcomes. Subheadings were added deductively (based on candidate theories) and inductively (as new themes emerged from the data). Data extraction was completed by the lead reviewer (HO), and a sample was double checked by a second reviewer (NC).

### Quality appraisal
Studies were not formally appraised at the data extraction stage, as would be the case in traditional systematic reviews. Instead, an assessment of 'rigour' was used to judge the credibility and trustworthiness of the evidence as it was integrated into the analysis and synthesis.[8 11] This assessment was not scored because weaker studies were still included, but it meant that methodological limitations were acknowledged and study findings were not overinterpreted. Table 3 shows the questions used to assess rigour. Quality appraisal was completed by the lead reviewer (HO) and double checked by second reviewers (NL/VM).

### Analysis and synthesis
This process involved gradual and iterative theory development, whereby evidence from primary studies was used to modify, refine and substantiate programme theories

**Table 3** Questions used to assess the rigour of primary studies

| # | Question |
|---|---|
| 1 | Are the study methods clearly reported (including study design, recruitment, data collection and analysis)? |
| 2 | Are the study methods appropriate to answer the research questions? |
| 3 | Are the sample characteristics reported to enable judgements about generalisability? |
| 4 | Are the study findings and conclusions supported by raw data? |
| 5 | Are the study limitations acknowledged and clearly reported? |

about how low-income pregnant women use Healthy Start vouchers, in what circumstances and why. Theories were constructed as explanatory CMO configurations, usually by starting with the outcome and working backwards to determine 'what caused it (the mechanism) and under what contexts was the mechanism triggered'.[12] The main focus of the analysis was searching for evidence to support and refute the proposed causal linkages between context, mechanisms and outcomes. A combination of evidence synthesis and realist analysis techniques was used:

1. Narrative synthesis of quantitative data on women's nutritional outcomes; meta-analysis was not appropriate due to heterogeneity of study designs and data collection methods (and was beyond the scope of this review).
2. Thematic synthesis of qualitative data, by creating codes and themes (as described under data extraction) and then 'going beyond' the interpretations of the original studies to generate new understandings or hypotheses.[13]
3. Creative theorising or 'retroduction' by the lead reviewer (HO) in collaboration with the review team and the stakeholder group. This involved in-depth reflection and discussions (throughout the review) about the underlying causes of outcome patterns, at the level of generative mechanisms and explanatory context. The data from included studies did not always provide such in-depth insights and explanations. Where individual extracts of data only supported part of the CMO configuration, it was necessary to make logical inferences about the complete causal pathways and explanations.[14]

## RESULTS
### Search results and study characteristics
A total of 908 records were identified through the two separate searches. After screening titles and abstracts, 88 records were obtained in full-text format. Fifty full-text articles were excluded based on the assessment of relevance (n=33) or because they were not primary studies (n=15) or the findings were duplicated (n=2). Therefore, 38 primary studies were included in this review: four UK studies on Healthy Start[6 7 15 16] and 34 US studies on WIC[17–50] (see online PRISMA Flow Diagram and online supplementary file 1).

### Identification of outcomes and supporting evidence
Two main outcome strands emerged during the theory development stage and were further substantiated using evidence from primary studies:

1. Women use vouchers to increase consumption of target foods—dietary improvements.
2. Women use vouchers to reduce food expenditure—financial assistance.

For the purposes of this review, we have assumed that strand 1 is the intended outcome of the programme and strand 2 is an unintended outcome. This was not explicit

in policy documentation, but there were references to dietary improvements which were thought to be achieved by enabling low-income women to access healthier foods and encouraging positive nutritional choices.[4] The included studies provided support for both outcome strands. The next section provides an overview of the available evidence on women's outcomes (intended and unintended) and highlights the relative contribution of evidence from Healthy Start and WIC studies. It also helps to illustrate how we worked backwards from outcomes to identify generative mechanisms and related aspects of context.

A total of 25 studies reported women's nutritional outcomes: three studies on Healthy Start and 22 studies on WIC. The Healthy Start studies reported perceived outcomes only; some women said they consumed more cow's milk, fruits and vegetables after receiving Healthy Start vouchers,[6 7 16] whereas other women said the vouchers 'freed up money to do other things' and 'helped them to manage better financially' (p 59).[7] The WIC studies were published between 1981 and 2015, but the most useful data was extracted from two studies comparing women's diets before and after the 2009 WIC revisions when the 'cash value vouchers' for fruits and vegetables were introduced (there was no allowance for fresh fruits and vegetables before 2009). A longitudinal study of African–American and Hispanic women from WIC clinics in Chicago (n=273) found significant dietary improvements for Hispanic mothers who reported consuming more fruit, more reduced fat milk less whole milk and less saturated fat (all p<0.05).[39] African–American mothers reported consuming less whole milk (p=0.02) but no other changes were statistically significant.[41] There were no sustained dietary improvements in either group compared with baseline at 18 months.[38] A cross-sectional study comparing two random samples of WIC participants in California (both 80% Hispanic) found that women assessed 6 months after the changes (n=2996) reported consuming significantly more whole grains, reduced-fat milk and vegetables and less whole milk compared with women assessed before the changes (n=3004) (all p<0.001).[49]

Five studies reported electronic sales data from WIC retailers (one large supermarket chain) in New England,

which showed that women's purchasing patterns shifted towards items provided in the WIC package after the 2009 revisions—fruits and vegetables, reduced-fat milk, brown rice, whole grain cereals and bread replaced less nutritious options.[18–22] One study (n=2137) showed that, while total spending on fruits and vegetables increased between 2009 and 2010 (p<0.001), up to 13% fewer purchases were made using non-WIC funds.[22] This implies that some women 'substituted' the method of payment, rather than using WIC to increase the amount of fruits and vegetables purchased. None of these studies reported sample characteristics such as ethnicity. Finally, a mixed-methods study of Hispanic and African–American pregnant women (n=313) found that two-thirds of women reported using WIC vouchers to reduce food expenditure.[33] The money they saved was used to purchase items for the unborn baby, other foods and for bills and emergencies.

These findings suggest that food vouchers may lead to dietary improvements for some, but not all women. This may be because some women use the vouchers to pay for foods they would previously have bought using cash. The WIC studies described above were not representative of ethnic groups in the UK, and the samples included mothers as well as pregnant women. However, these studies provide much needed evidence on the potential impact of food vouchers for low-income women, which was not available from the Healthy Start literature alone.

### Evidence-informed programme theories

This section presents three evidence-informed programme theories, which help to explain why different women receive the same Healthy Start vouchers and yet experience different outcomes because of variations in context and mechanisms. Figure 1 illustrates the key aspects of context, mechanisms and outcomes identified and the proposed causal pathways linking them together. These causal pathways are explained as CMO configurations and illustrated using quotations from included studies under each theory. Although low income is clearly an important aspect of context, we have not included it in our programme theories because it applies to all women who receive Healthy Start vouchers (apart from under 18s). As the programme theories explain, some women may achieve dietary improvements despite low income,

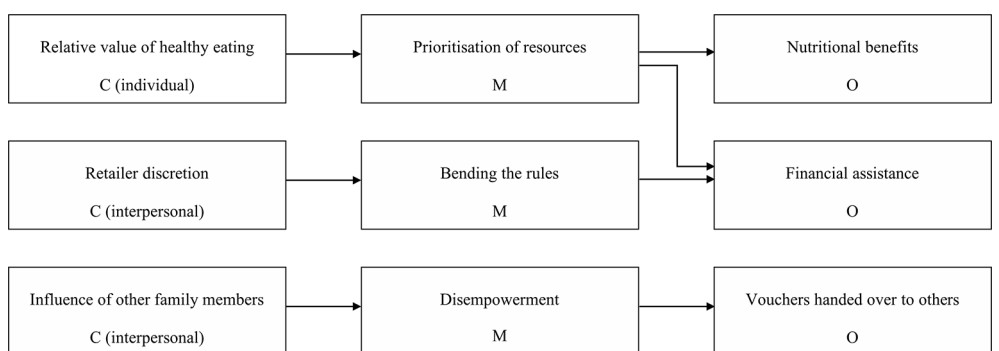

**Figure 1** Summary of programme theories about how low-income pregnant women use Healthy Start vouchers.

and other women may not—this divergence depends on other aspects of context.

### Prioritisation of resources

Women living on low income must constantly prioritise how they spend their money. Food vouchers may be considered as one part of the household resources and decisions must be made about how best to use the vouchers. A key aspect of individual level context is the 'relative value' of healthy eating (compared with other things women value), which can lead women to prioritise in different ways. Context is not static and women's values may change over time, so we propose that some women may fluctuate between the mechanisms (ways of prioritising) outlined in these two contrasting CMO configurations.

Women who value healthy eating and aspire to eat well during pregnancy (context)[47] are more likely to perceive Healthy Start vouchers as an opportunity to achieve health benefits for themselves and their unborn baby (mechanism).[6] The vouchers alleviate the financial barrier associated with healthy eating and make healthy foods seem more affordable (mechanism).[6] Therefore, women who value healthy eating are more likely to prioritise healthy eating (mechanism) and use Healthy Start vouchers to increase consumption of target foods—fruits and vegetables or cow's milk (outcome).[6]

'I have them at Asda when I do my shop, and I think how many vouchers I've got and I buy the veg that I have the vouchers for. I suppose if I didn't have the vouchers, I would just pick out the little things. I don't think if I didn't have the vouchers I'd buy half as much, no.' (Mother, UK; p 50)[6]

Alternatively, women may value healthy eating less than other things they want or need to spend money on, which are considered more important or urgent (context).[6 7] They are more likely to perceive Healthy Start vouchers as a way to save money, which can be redirected and prioritised in other ways (mechanism).[6 7] These women are more likely to use Healthy Start vouchers to deduct money from the shopping bill, with no increase in consumption of target foods (outcome).[6 7 33]

'Women are often in a dilemma about whether they should or shouldn't eat healthy foods because something else is needed more. Their own health and maybe the health of their younger children are on the back burner because something else is more pressing.' (Midwife, UK; page 35)[6]

### Bending the rules

The Healthy Start voucher exchange system relies on registered retailers to verify (visually) at the checkout that women have selected appropriate amounts of appropriate items—fruits and vegetables, plain cow's milk or infant formula. The vouchers are processed by swiping a bar code that subtracts the voucher value (£3.10) from the total. They are not electronically matched to specific food items. There is a reminder printed on each voucher about which foods may be purchased, along with a warning about prosecution, but the evidence suggests that some customers and retailers appear to disregard this information.

Retailers who are registered to accept Healthy Start vouchers have some discretion over how vigilantly they check what vouchers are spent on (context).[6 15] Women may put pressure on retailers to 'bend the rules' or make exceptions (mechanism).[6 15–17 32] Some retailers may decide to 'turn a blind eye' because they feel duty bound to help families in whatever ways they can (mechanism)[6] or because they prefer to avoid conflict (mechanisms).[16] This enables women to exchange the vouchers for alternative food or non-food items (outcomes).[6 16]

'But you have to realise that I get people coming in here, they are buying £1 pound of electricity every day. £1. That must run out after an hour. How do they live? And in the winter, it really does get very cold and they come in and ask me if they can use the voucher for electricity. What can I do? I can't see them living in the flat with young children, with no heating, it's so cold. So I do let them do that. They come in and show me their empty wallet and I have to believe them and I do sell gas and electricity for the voucher. You can report that back. I don't care, what can I do?' (Retailer, UK; p 69)[6]

### Disempowerment

Pregnant women (and later their young children) are the intended beneficiaries of Healthy Start, but some women may not be empowered to make decisions about how to use the vouchers themselves. The vouchers are posted to women at their home address, but there is no name printed on the actual vouchers and no identification is required at the checkout, so there is nothing to stop other people from using them. Regardless of what is bought with the vouchers, and who benefits, this would surely be considered an unintended outcome of the programme.

Women may not be empowered to make decisions about household resources or food shopping, such as pregnant teenagers who live with their parents (context)[6 47] or women who live in large, multigenerational households (context).[43] Women who are disempowered are more likely to hand over their Healthy Start vouchers to other family members (mechanism)[43] who then decide how they are used (outcome).[43]

'She (mom) makes most of the decisions. We get the same thing every time we go shopping.' (African–American mother living in multigenerational household, USA; p5).[43]

## DISCUSSION
### Statement of principal findings

This realist review identified two main outcome strands for low-income pregnant women using food vouchers from the Healthy Start programme: dietary improvements (intended) and financial assistance (unintended). Three evidence-informed programme theories were developed

and refined to explain how aspects of context (and mechanisms) may generate these outcomes: the 'relative value' of healthy eating (prioritisation of resources); retailer discretion (pressure to 'bend the rules'); the influence of other family members (disempowerment).

## Strengths of this review

This was the first study to use a realist, theory-driven approach to investigate how low-income pregnant women use Healthy Start vouchers, in what circumstances and why. The inclusion of relevant studies from a similar programme in the USA (WIC) provided insights and explanations beyond what was available from the Healthy Start literature. This was the first time that researchers have attempted to articulate, develop and test programme theories about how Healthy Start works. Our findings suggest that Healthy Start vouchers are not always used to achieve dietary improvements. Some low-income pregnant women may need to receive more support to increase the perceived value of healthy eating. Modifications to the voucher verification system would help to prevent the vouchers being used by other people and exchanged for alternative items. We hope this review will stimulate discussions about future evaluation needs and programme development.

## Limitations of this review

We were aware from the outset that some of the evidence from WIC studies would not be transferable to Healthy Start due to population and programme differences. Further evidence from the UK is needed to develop some of the other candidate theories we identified. This review focused on women's outcomes from the programme, and the aspects of context explored in our theories were individual (women's values and perceptions) and interpersonal (interactions with retailers and other family members).[8] We did not find sufficient evidence to link our CMO configurations with sociodemographic and cultural characteristics, such as which groups of women are more/less likely to value healthy eating. Future reviews in this area could include wider evidence on voucher programmes to provide insights about how the mechanisms we have identified might operate in different contexts and different programmes. This was beyond the scope of this review due to time and resource limitations. Finally, we did not explore theories relating to Healthy Start vitamins, women's decisions about infant feeding or children's outcomes. These would all be worthy areas for realist investigations.

## Comparisons with existing literature and theories

This study builds on two previous evaluations of Healthy Start, which highlighted different ways of using the food vouchers: some women used the vouchers to access healthy foods that they otherwise could not afford, whereas other women used them to save money on foods they had already planned to buy and reallocated the money for other things.[6 7] Our realist review has shown

how 'substitution effects' (ie, using the vouchers as an alternative method of payment) may reduce the potential impact on women's nutritional outcomes and some women may experience no dietary improvements at all. In addition, we have identified aspects of context and causal mechanisms that are likely to be important in determining outcome patterns for low-income pregnant women.

Our first programme theory relates to the economics of decision making. If women value healthy eating and want to do everything they can to give their baby the best possible start in life, these beliefs and motivations will influence the decision-making process when it comes to using the vouchers. However, other factors will also influence the decision-making process and women must consider whether additional fruits and vegetables (or cow's milk) are the most important things they need. Frick considered the everyday economic analyses that take place at family level in relation to infant and young child feeding, whereby mothers and other family members must decide how to allocate financial resources, weighing up food choices and nutrition against a range of other considerations.[51] He described how societal and individual values influence these trade-offs between nutrition and other priorities. Decisions about how to use Healthy Start vouchers are subject to similar trade-offs through the mechanism of prioritisation. Attree found that low-income women 'strategically adjust' to poverty by prioritising or 'juggling' what they spend money on.[52] Food may be ranked below other basic needs such as rent and household bills, with more flexibility to cut back on healthy items like fruits and vegetables. Therefore, Healthy Start may be seen as a way to manage financially by reducing food expenditure. The programme provides additional resources to (ideally) enable low-income pregnant women to improve their diets, but only women who value healthy eating (and the associated health benefits for mother and child) are likely to use the vouchers in this intended way. A recent taxonomy of behaviour change techniques defined 'incentives' as rewarding and contingent on behaviour change.[53] Healthy Start does not fit this definition and should not be assumed to encourage healthy choices for all beneficiaries.

Our second programme theory relates to retailers who misuse the Healthy Start programme by allowing women to exchange vouchers for alternative items. It is presented under the context of retailer discretion, which highlights weaknesses in the system, but this theory also relates to the context of women who value other things above healthy eating. The evidence suggests that some retailers may bend the rules because they feel they are acting in the best interests of the customer. This is similar to 'responsible subversion' identified among health professionals who admitted to bending or breaking the rules for what they perceived to be patient benefits, despite contravening evidence-based practice guidance.[54] However, there may be other (unscrupulous) reasons why retailers bend the rules and

further evidence is needed to develop and explore this programme theory more fully.

Finally, our third programme theory relates to women who may not be empowered to decide how to use their Healthy Start vouchers. Their choices may be heavily influenced (or constrained) by significant others, for example a partner, mother or mother-in-law, who may take charge of food shopping and allocation of household resources. Similar issues have been identified in relation to decisions about infant feeding: women are surrounded by networks of people who participate in decision making, so they may be unable to exercise their 'right to choose' despite knowing what the options are and possessing their own opinions.[55] This may be particularly the case in communities where there are high levels of interdependence within the extended family network.

Healthy Start is dependent on individual agency to achieve dietary improvements, in contrast with other types of nutrition interventions that can be said to be less dependent on individual agency (such as food fortification). Our evidence-informed programme theories illustrate how aspects of context may enable or constrain women's agency. A recent paper by the Centre for Diet and Activity Research considered the role of individual agency in public health interventions, concluding that 'low agency' interventions are more likely to be effective and equitable by reducing the need for individual decisions.[56] Food vouchers for free fruits and vegetables were positioned in the middle of a continuum of the degree of agency required to benefit from the intervention. This review highlights some ways in which the level of agency required could be reduced in the Healthy Start programme, such as by 'tightening up' the system for verifying who uses the vouchers and what they are exchanged for. However, agency cannot be eliminated from food voucher programmes and therefore this review contributes to ongoing debates about how public health interventions should be designed to maximise outcome effectiveness.

Agency is synonymous with realist mechanisms (the reasoning and reactions of individuals in response to the resources offered by the programme), and this review illustrates the contribution of realist methodology to understanding the differential impacts of public health interventions or programmes.

## CONCLUSION

This realist review suggests that some low-income pregnant women use Healthy Start vouchers to increase their consumption of fruits and vegetables and plain cow's milk (intended outcome: dietary improvements), whereas other women use them to reduce food expenditure and save money for other things (unintended outcome: financial assistance). We have identified some aspects of context (the 'relative value' of healthy eating, retailer discretion and the influence of other family members) and mechanisms (prioritisation of resources,

pressure to 'bend the rules' and disempowerment) that are likely to be important in determining these outcomes. Further evidence is needed to understand how low-income pregnant women could be better supported to prioritise healthy eating and use Healthy Start vouchers to improve their diets during pregnancy— in particular to buy more fruits and vegetables. This may include ways of 'tightening up' the programme to reduce the amount of agency required but also considering ways in which women may be supported to become more empowered to choose.

**Contributors** HO was the lead reviewer, while other authors completed the second reviewer tasks (as indicated in the Methods). All authors contributed to the development of programme theories through regular discussions. HO drafted the initial manuscript and all authors contributed to the final manuscript.

**Competing interests** None declared.

**Ethics approval** University of Central Lancashire Science, Technology, Engineering, Medicine and Health (STEMH) Ethics Committee.

**Provenance and peer review** Not commissioned; externally peer reviewed

**Data sharing statement** No additional data are available.

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
