## [Reviewer comments · BMJ Open]

ARTICLE DETAILS

TITLE (PROVISIONAL)	A realist review to explore how low-income pregnant women use food vouchers from the UK's Healthy Start programme
AUTHORS	Ohly, Heather; Crossland, Nicola; Dykes, Fiona; Lowe, Nicola; Hall-Moran, Victoria

VERSION 1 - REVIEW

REVIEWER	Helen Cheyne NMAHP Research Unit, University of Stirling, UK
REVIEW RETURNED	09-Sep-2016

GENERAL COMMENTS	Thank you for giving me the opportunity to review this interesting paper. The topic is interesting and important and the authors are to be complimented on their use of a realist synthesis approach which is appropriate to address the issues concerned. Realist synthesis methods are becoming more popular and although there is quite a lot of guidance for how to conduct a realist synthesis there are very few good examples to guide how best to report this complicated review method. I hope that the following comments will be helpful in revision of this paper. I think that realist synthesis is an appropriate method for this review and it is clear that the authors do understand and follow the main steps required. However some of this unfolds through the paper and I think that, given this is a complicated method and significantly different from a more standard review, there should be some more explanation / justification of the review approach and key stages in the introduction. The introduction makes the statement that there were a limited number of studies on Healthy Start but there is nothing in the introduction to explain how they came to this conclusion. Was it on the basis of a preliminary review of literature? A key element of realist synthesis is that the findings are presented in relation to the research questions that arise from the original programme theory. An important first stage is theory development, developing of a 'folk theory' of why and how an intervention or programme is anticipated to work. The authors do clearly say this in the introduction. They do explain how programme theories were developed in the methods section, and appear to have used appropriate methods to do this part of the work. They make reference to the programme theory but it/they are not included explicitly in the paper. I think it would improve clarity if the programme theories that were developed as part of the review were presented, perhaps before the research questions, as these arise from the theories. Methods As above the programme theory development section is a bit unclear, for example the authors say – the programme has an
--

implicit theory that exists..so they used a bottom up approach to theory development, with candidate theories. While I think I know what they mean, I feel that this could be more clearly explained. Possibly an overview of realist synthesis methods in the introduction might help.

Similarly in relation to the search strategy I feel that a brief explanation of the ways in which a realist synthesis search strategy differs from a more standard review would be helpful. I agree with their questions and assessment of relevance although I am not sure what question 3 means.

Data extraction – subheadings were added deductively based on ‘early theories’, I feel we need to know what these early theories were. Again, I think that all of the methods would be clearer if the programme theory stage was presented first.

The analysis section. I understand the first point – that evidence drawn substantiates or refutes initial programme theories. However the second point – theories constructed as explanatory CMO configurations – starting with the outcome and working backwards is confusing. I wonder if an example would help here?

The third point – the focus was on search for evidence to support proposed causal linkages. Should this be to support or refute proposed causal linkages?

The section explaining the combination of evidence synthesis and realist analysis is quite difficult to follow.

Results

The section that says the decision was made to focus on the final outcomes of low income women as first beneficiaries of the programme is not well explained. The authors say that the decision was made during the theory development stage. I feel that this was probably a sensible decision as realist synthesis can produce a wealth of material and several other publications have also chosen to present only one part of the findings. The problem is that as the findings of the theory development phase are not presented the wider picture is not clear and it could seem as if a largely resource based decision was made to exclude important aspects such as how women access the programme. Given the importance placed on context in realist approaches this would seem odd. Presenting the findings of the theory development stage and then going on to take one part of it in this paper (as the authors have done) would make it clearer.

There are several places in the results section where reference is made to the findings of the theory development phase which are not particularly understandable in the absence of a report on this stage. For example – two main outcome strands emerged during the theory development phase 1 is presumed to be intended and 2 unintended, this seems a bit unsubstantiated at present.

I am not clear about the purpose of the first part of the outcomes section is. I think it is the narrative synthesis of the quantitative data but I am not sure of its place in the realist synthesis – I think its inclusion needs some more explanation.

The section on CMO configurations is much clearer and is presented well, this much more clearly the expected format of a realist synthesis method.

Overall, I think that the paper would be considerably improved by a section providing an overview of realist synthesis methods in the introduction and a section presenting the programme theory development phase findings.

REVIEWER	Rebecca Hardwick University of Exeter Medical School, United Kingdom
REVIEW RETURNED	12-Sep-2016

GENERAL COMMENTS	Thank you for the opportunity to review this paper. I found it an engaging and clear read. My comments are focussed on clarity and readability. Abstract – no comments. Background p4. First paragraph, second sentence, I was unclear what the relevance of the words in brackets were, and they were unclear to me I understand that women get food vouchers worth £3:10 during pregnancy, but do the words in brackets mean they get £6.20 if they already have a child under one year, so the £6:20 is in addition to the £3.10? Just needs clarifying. Line 13 - who do the retailers need to be registered with? Lines 13-21 - I think these few sentences need rephrasing - I think they are the reason for the review only focussing on the food component but this could be stated more clearly. The authors might also like to reflect on this in the discussion/areas for future research. Line 55-56 - authors could consider mentioning how "analysis" is the process through which ideas or hypotheses are transformed into evidence-based explanations. p5. Authors could state more firmly why it is possible to use findings from the WIC, US studies in this review of a UK programme. A brief discussion of the function of abstraction, and generative mechanisms in realist research, and references to how other published realist research has used 'sister' programmes from the US alongside UK programmes would be helpful. The authors could consider whether they need to also highlight the literature on behaviour change/incentive theory (which is the focus of the discussion) in this background section. Methods Clear, and well explained. Impressive! p7 Lines 13-14. It would be useful to know how the questions in table 2 were developed – was it through consultation with the stakeholder group? Through initial scoping of the literature? I think the table is a good example of how relevance can be assessed and it would be good to know how it was designed. p9 Line 3 I don't think the inverted commas around "stage" are necessary. In fact, potentially "stage" could be deleted altogether? Lines 8-12 "working backwards" – I think this needs further elaboration as to a novice reader, it might be unclear what is meant. Lines 16-33 is one of the clearest accounts of the process of evidence synthesis I've seen in a realist review. Well done. Results p10 Lines 14-20, this part discussed "the two main outcome strands which emerged during theory development. That women use vouchers to increase consumption of target foods – nutritional benefits, and that women use vouchers to reduce food expenditure – financial assistance." This makes sense to me. However, lines 30 onwards talk about "nutritional outcomes", which leads me to think about the actual nutritional outcomes – ie. Increases in calcium or iron. Which might be the ultimate intention, but isn't the outcome under review. Adding in "benefits" between 'nutritional' and 'outcomes' would clarify this for me. Over the page, there is another
--

slightly different iteration of the nutritional outcome, as "dietary improvements" (line24/25). It would benefit the paper if one term could be used to describe the nutritional outcome, and to have that outcome clearly defined at the start of the section.

p10, lines 23-26 – It would be helpful to justify why, although not explicit in policy, it was decided that strand 2 was unintended, and strand 1 intended. I wonder whether the financial assistance outcome is an interim outcome (or mechanism) leading to nutritional benefits.... Are they mutually exclusive? I think this paragraph needs a little more explanation and detail.

The two studies referred to later in that paragraph – were they self-reported/ how was data collected? This may have a bearing on the rigour of the findings. Can the p value of significance for the Whaley et al 2012 study be reported too?

I am torn with whether to offer further comment on the remainder of the results. On the one hand, they read simply and clearly, are well argued and succinct. All good things. On the other, I find myself wanting a bit more depth and detail of the studies which inform the argument. I wonder whether more description of the different studies would be helpful, both the kind of research and the context of the research. For example, p12, lines 11-22 it would be really interesting to know more about studies 45 and 11 – I find myself wanting more depth here. Why was it that women valued healthy eating more? How was that judged? Was it something the researchers conducting the study concluded, or was it explicitly referred to by the women themselves?

p12, Line 45-48 – "[...] some customer and retailers appear to disregard this information" I'm interested in where this knowledge comes from – is it from a single study or evaluation? As the programme theory on bending the rules is based on it, it would be helpful to know how you know.

Discussion

P13, line 41/42 – not sure you need commas around evidence-informed (ditto for p 11 40-41). The programme theories were indeed informed by the evidence. Also, I think this sentence needs to reflect that programme theories were developed and refined, rather than just proposed. A hallmark of realist work is not just the building of programme theory, but subjecting those theories to the literature to refine them. Which you did.

Strengths of the review

I wonder if it is possible to reflect on what the study has contributed to knowledge of how voucher schemes 'work'? What is included in this section is all true – novel methodological approach, use of US data, developing programme theories, but I feel there is more that could be said about what the findings mean, above and beyond a simple 'pass/fail' judgement on the use of vouchers... Perhaps a sentence to elaborate on how it can stimulate discussion in the field? So what it might mean for practice, if a decision maker was intending on setting up a voucher scheme similar to this – what would they need to know/bear in mind? (I'm not sure whether that kind of discussion sits here or elsewhere in the discussion section.)

Limitations of the review

It is highlighted that the focus of the review has been at the interpersonal and individual contextual level – it would be helpful to readers to have some knowledge of this as a realist construct earlier in the paper, perhaps in the methods section where the discussion of not looking at sociodemographic (low-income) factors is mentioned? Also, it occurs to me that the justifications for not looking at this contextual level are given at one point in the review as

"insufficient evidence", and at another point in the review as "irrelevant" (almost), as women's eligibility for the vouchers is predicated on their being on a low-income already. I think this needs resolving.

Could this section also include how in the confines of this review (e.g. time available) it was not possible to look at how the mechanisms identified here operate in other kinds of programmes which use financial incentives (and therefore other contexts), but how doing so would test the mechanisms further, leading to more robust programme theory and understanding about how "vouchers" are used more generally?

Comparisons with existing literature and theories

I think the first paragraph need to be clearer in the point it is making.

I am curious to know broadly how do the review findings illuminate/develop/dismiss incentive theory and behaviour change?

In particular, this is the first explicit mention of financial incentives, so does the link need to be made earlier that the financial assistance that vouchers provide could be construed as a financial incentive towards healthy eating? In effect, an outcome which turns into a mechanism? Was this discussed at the theory building stage?

I also wonder whether reference to other realist literature which has looked into how the mechanisms identified in this review relate to behaviour change, and how they may be relevant. Unfortunately I can't think of any off the top of my head, but an online search/question to RAMESES community may be fruitful.

(<https://www.jiscmail.ac.uk/cgi-bin/webadmin?A0=RAMESES>)

Conclusions

The review has found some important contextual factors which have an effect on how financial incentives work and I think it would be good to extrapolate from the review findings what this might mean for decision and policy makers.

Figure 1. Authors could highlight the 'level' at which these contexts operate (individual/ interpersonal).

Additional points for authors to consider:

Adherence to BMJ Open Instructions to Authors:

In my opinion, the article adheres to BMJ Open Instructions to Authors.

Adherence to RAMESES publication standards – a few suggestions below:

5 Changes in the review process Any changes made to the review process that was initially planned should be briefly described and justified. – I don't think this hasn't been addressed so if no changes were made to the review process, then this should be reported.

13 Document characteristics Provide information on the characteristics of the documents included in the review.

Authors could consider including a table which provides readers with the document characteristics. RAMESES publication guidelines suggest the following as potential headings: Examples of possibly relevant characteristics of documents that may be worth reporting include, where applicable: full citation, country of origin, study design, summary of key main findings, use made of document in the synthesis and relationship of documents to each other (for example, there may be more than one document reporting on an intervention).

VERSION 1 – AUTHOR RESPONSE

Responses to Reviewer: 1

Reviewer Name: Helen Cheyne

Institution and Country: NMAHP Research Unit, University of Stirling, UK

Reviewer comment: Thank you for giving me the opportunity to review this interesting paper. The topic is interesting and important and the authors are to be complimented on their use of a realist synthesis approach which is appropriate to address the issues concerned. Realist synthesis methods are becoming more popular and although there is quite a lot of guidance for how to conduct a realist synthesis there are very few good examples to guide how best to report this complicated review method. I hope that the following comments will be helpful in revision of this paper.

Author response: Thank you Helen. We have done our best to respond to your detailed comments. In particular, we have added more information on the realist approach (Background) and a new section of our findings from the theory development stage (Results).

Reviewer comment: I think that realist synthesis is an appropriate method for this review and it is clear that the authors do understand and follow the main steps required. However, some of this unfolds through the paper and I think that, given this is a complicated method and significantly different from a more standard review, there should be some more explanation / justification of the review approach and key stages in the introduction.

Author response: We have added a more substantial introduction to the realist approach, realist synthesis and programme theories into the Background section. We hope this helps to set the scene before we describe our review methods and findings.

Reviewer comment: The introduction makes the statement that there were a limited number of studies on Healthy Start but there is nothing in the introduction to explain how they came to this conclusion. Was it on the basis of a preliminary review of literature?

Author response: Yes, we did a preliminary search and this has been clarified in the introduction.

Reviewer comment: A key element of realist synthesis is that the findings are presented in relation to the research questions that arise from the original programme theory. An important first stage is theory development, developing of a 'folk theory' of why and how an intervention or programme is anticipated to work. The authors do clearly say this in the introduction. They do explain how programme theories were developed in the methods section, and appear to have used appropriate methods to do this part of the work. They make reference to the programme theory but it/they are not included explicitly in the paper. I think it would improve clarity if the programme theories that were developed as part of the review were presented, perhaps before the research questions, as these arise from the theories.

Author response: We have added more detail in the Results section on our findings from the theory development stage. This explains the broader scope of our candidate programme theories (including examples of factors/issues we considered) and then how we decided which ones to test. In the Background section, the information we have added about the realist approach leads into our review questions, which we have reworded to be clearer and more explicitly realist.

Methods

Reviewer comment: As above the programme theory development section is a bit unclear, for example the authors say – the programme has an implicit theory that exists...so they used a bottom

up approach to theory development, with candidate theories. While I think I know what they mean, I feel that this could be more clearly explained. Possibly an overview of realist synthesis methods in the introduction might help.

Author response: We have added an overview of the realist approach and realist synthesis methods (Background) and another sentence to clarify what we mean by 'bottom up' theory development (Methods). We hope this is all much clearer now.

Reviewer comment: Similarly, in relation to the search strategy I feel that a brief explanation of the ways in which a realist synthesis search strategy differs from a more standard review would be helpful. I agree with their questions and assessment of relevance although I am not sure what question 3 means.

Author response: We have added the following sentence into the Background, which helps to explain our search strategy: "Evidence may be obtained from studies of the programme itself, or more widely from similar programmes or programmes that are thought to work in similar ways." We have clarified question 3 in Table 2.

Reviewer comment: Data extraction – subheadings were added deductively based on 'early theories', I feel we need to know what these early theories were. Again, I think that all of the methods would be clearer if the programme theory stage was presented first.

Author response: We have added more detail on the candidate theories we developed, but we feel this belongs in the Results not the Methods. We hope the additional Background information on realist synthesis will help the reader to understand what this means before they reach the Methods.

Reviewer comment: The analysis section. I understand the first point – that evidence drawn substantiates or refutes initial programme theories. However, the second point – theories constructed as explanatory CMO configurations – starting with the outcome and working backwards is confusing. I wonder if an example would help here?

Author response: We have added a quotation from Geoff Wong's paper to clarify how outcomes were related backwards to mechanisms and contexts.

Reviewer comment: The third point – the focus was on search for evidence to support proposed causal linkages. Should this be to support or refute proposed causal linkages?

Author response: Yes, thank you, we have added this.

Reviewer comment: The section explaining the combination of evidence synthesis and realist analysis is quite difficult to follow.

Author response: We have provided clearer explanation of the terms used in point 3 because this is likely to be the method that some readers are least familiar with. The other methods described in 1 and 2 are commonly used in evidence synthesis.

Results

Reviewer comment: The section that says the decision was made to focus on the final outcomes of low income women as first beneficiaries of the programme is not well explained. The authors say that the decision was made during the theory development stage. I feel that this was probably a sensible decision as realist synthesis can produce a wealth of material and several other publications have

also chosen to present only one part of the findings. The problem is that as the findings of the theory development phase are not presented the wider picture is not clear and it could seem as if a largely resource based decision was made to exclude important aspects such as how women access the programme. Given the importance placed on context in realist approaches this would seem odd. Presenting the findings of the theory development stage and then going on to take one part of it in this paper (as the authors have done) would make it clearer.

Author response: We have added more details on our findings from the theory development stage, as suggested. This includes an overview of the stages of the programme pathway we considered and examples of the factors we theorised. Then we have explained why we prioritised the last stage of the programme pathway for testing. Thank you for this comment, which we agree has made the Results section much clearer.

Reviewer comment: There are several places in the results section where reference is made to the findings of the theory development phase which are not particularly understandable in the absence of a report on this stage. For example – two main outcome strands emerged during the theory development phase 1 is presumed to be intended and 2 unintended, this seems a bit unsubstantiated at present.

Author response: This should be clearer now (see above responses).

Reviewer comment: I am not clear about the purpose of the first part of the outcomes section is. I think it is the narrative synthesis of the quantitative data but I am not sure of its place in the realist synthesis – I think its inclusion needs some more explanation.

Author response: We have clarified the reasons why we included this section on outcomes: “The next section provides an overview of the available evidence on women’s outcomes (intended and unintended) and highlights the relative contribution of evidence from Healthy Start and WIC studies. It also helps to illustrate how we worked backwards from outcomes to identify generative mechanisms and related aspects of context.”

Reviewer comment: The section on CMO configurations is much clearer and is presented well, this much more clearly the expected format of a realist synthesis method.

Author response: Thank you.

Reviewer comment: Overall, I think that the paper would be considerably improved by a section providing an overview of realist synthesis methods in the introduction and a section presenting the programme theory development phase findings.

Author response: Thanks again for your constructive comments. We agree that these changes have improved the paper considerably.

Responses to Reviewer: 2

Reviewer Name: Rebecca Hardwick

Institution and Country: University of Exeter Medical School, United Kingdom

Reviewer comment: Thank you for the opportunity to review this paper. I found it an engaging and clear read. My comments are focussed on clarity and readability.

Author response: Thank you Rebecca. We have done our best to respond to your detailed comments.

In particular, we have added more information on the realist approach (Background), clarified our descriptions of the outcome strands (Results), added raw data to support the evidence-informed programme theories (Results) and emphasised the contribution/implications of our review (Discussion).

Reviewer comment: Abstract – no comments.

Author response: No response needed.

Background

p4.

Reviewer comment: First paragraph, second sentence, I was unclear what the relevance of the words in brackets were, and they were unclear to me I understand that women get food vouchers worth £3.10 during pregnancy, but do the words in brackets mean they get £6.20 if they already have a child under one year, so the £6.20 is in addition to the £3.10? Just needs clarifying.

Author response: These amounts relate to how the voucher value changes from pregnancy to motherhood. We hope the wording of this sentence is now clearer.

Reviewer comment: Line 13 - who do the retailers need to be registered with?

Author response: A little more detail has been added to clarify this.

Reviewer comment: Lines 13-21 - I think these few sentences need rephrasing - I think they are the reason for the review only focussing on the food component but this could be stated more clearly. The authors might also like to reflect on this in the discussion/areas for future research.

Author response: Yes, it was the main reason for focusing on the food vouchers. We have adjusted the wording in the first paragraph to make this clearer. We have also added more explicit suggestions for future research in the Discussion as suggested.

Reviewer comment: Line 55-56 - authors could consider mentioning how "analysis" is the process through which ideas or hypotheses are transformed into evidence-based explanations.

Author response: This whole section on the realist approach has been expanded and improved, and we have emphasised the gradual and iterative nature of the review process.

p5.

Reviewer comment: Authors could state more firmly why it is possible to use findings from the WIC, US studies in this review of a UK programme. A brief discussion of the function of abstraction, and generative mechanisms in realist research, and references to how other published realist research has used 'sister' programmes from the US alongside UK programmes would be helpful.

Author response: The introduction has been expanded to include a more substantial introduction to the realist approach, including the use of evidence from similar programmes that are thought to work in similar ways. This comes ahead of our justification for including WIC studies. We have also added more detail on different levels of abstraction and better descriptions of context and generative mechanisms. We hope you agree that the whole paper flows better since adding this.

Reviewer comment: The authors could consider whether they need to also highlight the literature on behaviour change/incentive theory (which is the focus of the discussion) in this background section.

Author response: We have added a sentence to introduce the idea that Healthy Start vouchers may have been conceived as a financial incentive for dietary improvements. However, we feel the place to discuss this literature in more depth is the Discussion.

Methods

Reviewer comment: Clear, and well explained. Impressive!

Author response: Thank you, this is great to hear!

Reviewer comment: p7 Lines 13-14. It would be useful to know how the questions in table 2 were developed – was it through consultation with the stakeholder group? Through initial scoping of the literature? I think the table is a good example of how relevance can be assessed and it would be good to know how it was designed.

Author response: We have added a sentence to clarify how and when this was done.

p9

Reviewer comment: Line 3 I don't think the inverted commas around "stage" are necessary. In fact, potentially "stage" could be deleted altogether?

Author response: This has now been removed.

Reviewer comment: Lines 8-12 "working backwards" – I think this needs further elaboration as to a novice reader, it might be unclear what is meant.

Author response: We have added a quotation from Geoff Wong's paper to clarify how outcomes can be related backwards to mechanisms and contexts.

Reviewer comment: Lines 16-33 is one of the clearest accounts of the process of evidence synthesis I've seen in a realist review. Well done.

Author response: Thank you again!

Results

Reviewer comment: p10 Lines 14-20, this part discussed "the two main outcome strands which emerged during theory development. That women use vouchers to increase consumption of target foods – nutritional benefits, and that women use vouchers to reduce food expenditure – financial assistance." This makes sense to me. However, lines 30 onwards talk about "nutritional outcomes", which leads me to think about the actual nutritional outcomes – i.e. Increases in calcium or iron. Which might be the ultimate intention, but isn't the outcome under review. Adding in "benefits" between 'nutritional' and 'outcomes' would clarify this for me. Over the page, there is another slightly different iteration of the nutritional outcome, as "dietary improvements" (line24/25). It would benefit the paper if one term could be used to describe the nutritional outcome, and to have that outcome clearly defined at the start of the section.

Author response: We have changed our terminology throughout the review to refer to dietary improvements in outcome strand 1. We agree this is clearer and more consistent. More generally we still refer to women's nutritional outcomes (which can include food and nutrient intakes as you have understood) because this phrase does not impose the direction of effect.

Reviewer comment: p10, lines 23-26 – It would be helpful to justify why, although not explicit in policy, it was decided that strand 2 was unintended, and strand 1 intended. I wonder whether the financial assistance outcome is an interim outcome (or mechanism) leading to nutritional benefits.... Are they mutually exclusive? I think this paragraph needs a little more explanation and detail.

Author response: We stated that the distinction between intended and unintended outcomes was our assumption and was not explicit in the policy documentation. We have added: "...but there were references to dietary improvements, which were thought to be achieved by enabling low-income women to access healthier foods and encouraging positive nutritional choices." We have also added this sentence under theory 1 for clarification: "Context is not static and women's values may change over time, so some women may fluctuate between the mechanisms (ways of prioritising) outlined in these two contrasting CMO configurations."

Reviewer comment: The two studies referred to later in that paragraph – were they self-reported/ how was data collected? This may have a bearing on the rigour of the findings. Can the p value of significance for the Whaley et al 2012 study be reported too?

Author response: We have added p values and clarified that dietary assessment was self-reported.

Reviewer comment: I am torn with whether to offer further comment on the remainder of the results. On the one hand, they read simply and clearly, are well argued and succinct. All good things. On the other, I find myself wanting a bit more depth and detail of the studies which inform the argument. I wonder whether more description of the different studies would be helpful, both the kind of research and the context of the research. For example, p12, lines 11-22 it would be really interesting to know more about studies 45 and 11 – I find myself wanting more depth here. Why was it that women valued healthy eating more? How was that judged? Was it something the researchers conducting the study concluded, or was it explicitly referred to by the women themselves?

Author response: Due to word count limitations, we were unable to go into detail about the studies that substantiated each theory. However, we have added some illustrative quotes to help explain how the evidence support the CMO. This is a similar format to the findings presented in other realist reviews published in BMJ Open. As explained in the Methods, we did not find evidence linking the entire CMO and to some extent we inferred the linkages: "Where individual extracts of data only supported part of the CMO configuration, it was necessary to make logical inferences about the complete causal pathways and explanations."

Reviewer comment: p12, Line 45-48 – "[...] some customer and retailers appear to disregard this information" I'm interested in where this knowledge comes from – is it from a single study or evaluation? As the programme theory on bending the rules is based on it, it would be helpful to know how you know.

Author response: This statement relates to the mechanisms in theory 3 (women bending the rules and retailers turning a blind eye) which suggests they are disregarding the rules of the programme. We have added a quote that illustrates this intent.

Discussion

Reviewer comment: P13, line 41/42 – not sure you need commas around evidence-informed (ditto for p 11 40-41). The programme theories were indeed informed by the evidence. Also, I think this sentence needs to reflect that programme theories were developed and refined, rather than just proposed. A hallmark of realist work is not just the building of programme theory, but subjecting those

theories to the literature to refine them. Which you did.

Author response: We have removed the inverted commas and adjusted the wording as suggested. We agree this reflects what we actually did with more confidence.

Strengths of the review

Reviewer comment: I wonder if it is possible to reflect on what the study has contributed to knowledge of how voucher schemes 'work'? What is included in this section is all true – novel methodological approach, use of US data, developing programme theories, but I feel there is more that could be said about what the findings mean, above and beyond a simple 'pass/fail' judgement on the use of vouchers... Perhaps a sentence to elaborate on how it can stimulate discussion in the field? So what it might mean for practice, if a decision maker was intending on setting up a voucher scheme similar to this – what would they need to know/bear in mind? (I'm not sure whether that kind of discussion sits here or elsewhere in the discussion section.)

Author response: We have added the following sentences to sum up what our findings mean for practice: "Our findings suggest that Healthy Start vouchers are not always used to achieve dietary improvements. Some pregnant women may need to receive more support to increase the perceived value/importance of healthy eating. Modifications to the voucher verification system would help to prevent the vouchers being used by other people and exchanged for alternative items. We hope this review will stimulate discussions about future evaluation needs and programme development."

Limitations of the review

Reviewer comment: It is highlighted that the focus of the review has been at the interpersonal and individual contextual level – it would be helpful to readers to have some knowledge of this as a realist construct earlier in the paper, perhaps in the methods section where the discussion of not looking at sociodemographic (low-income) factors is mentioned? Also, it occurs to me that the justifications for not looking at this contextual level are given at one point in the review as "insufficient evidence", and at another point in the review as "irrelevant" (almost), as women's eligibility for the vouchers is predicated on their being on a low-income already. I think this needs resolving.

Author response: We have added more information on the realist approach into the Background and this includes the four levels of context. It is not until we present our programme theories in the Results that we explain and justify which aspects of context we explored. We did not mean to infer that income is not relevant and we have tried to clarify this sentence. Later, where we refer to insufficient evidence in the Limitations, this relates to women's "sociodemographic and cultural characteristics" (not just income) and the fact that our theories do not explain which women are more/less likely to value healthy eating.

Reviewer comment: Could this section also include how in the confines of this review (e.g. time available) it was not possible to look at how the mechanisms identified here operate in other kinds of programmes which use financial incentives (and therefore other contexts), but how doing so would test the mechanisms further, leading to more robust programme theory and understanding about how "vouchers" are used more generally?

Author response: We have added this sentence: "Future reviews in this area could include wider evidence on voucher programmes to provide insights about how the mechanisms we have identified might operate in different contexts and different programmes. This was beyond the scope of this review due to time and resource limitations."

Comparisons with existing literature and theories

Reviewer comment: I think the first paragraph need to be clearer in the point it is making. I am curious to know broadly how do the review findings illuminate/develop/dismiss incentive theory and behaviour change? In particular, this is the first explicit mention of financial incentives, so does the link need to be made earlier that the financial assistance that vouchers provide could be construed as a financial incentive towards healthy eating? In effect, an outcome which turns into a mechanism? Was this discussed at the theory building stage?

Author response: Thank you for this comment. We have clarified (both here and earlier in the paper) that that Healthy Start vouchers may have been conceived as a financial incentive for dietary improvements. In this section, we have improved our comparisons between Healthy Start and the various existing definitions and theories. We have emphasised the importance of context in determining whether the monetary value of the vouchers is perceived as an opportunity to achieve health benefits or a way to save money.

Reviewer comment: I also wonder whether reference to other realist literature which has looked into how the mechanisms identified in this review relate to behaviour change, and how they may be relevant. Unfortunately, I can't think of any off the top of my head, but an online search/ question to RAMESES community may be fruitful. (<https://www.jiscmail.ac.uk/cgi-bin/webadmin?A0=RAMESES>)

Author response: As far as we are aware, after several searches and discussions with experts in the field of realist synthesis/evaluation, this is the first realist review to explore non-conditional financial incentives for dietary improvement such as food voucher programmes.

Conclusions

Reviewer comment: The review has found some important contextual factors which have an effect on how financial incentives work and I think it would be good to extrapolate from the review findings what this might mean for decision and policy makers.

Author response: We have added the following sentence: "Targeted support may be needed to help some low-income pregnant women to use Healthy Start vouchers to improve their diets during pregnancy."

Reviewer comment: Figure 1. Authors could highlight the 'level' at which these contexts operate (individual/ interpersonal).

Author response: Thank you, we have added this into Figure 1.

Additional points for authors to consider:

Adherence to BMJ Open Instructions to Authors:

Reviewer comment: In my opinion, the article adheres to BMJ Open Instructions to Authors.

Author response: No response needed.

Adherence to RAMESES publication standards – a few suggestions below:

Reviewer comment: 5 Changes in the review process Any changes made to the review process that was initially planned should be briefly described and justified. – I don't think this hasn't been

addressed so if no changes were made to the review process, then this should be reported.

Author response: This has been clarified.

Reviewer comment: 13 Document characteristics Provide information on the characteristics of the documents included in the review. – Authors could consider including a table which provides readers with the document characteristics. RAMESES publication guidelines suggest the following as potential headings: Examples of possibly relevant characteristics of documents that may be worth reporting include, where applicable: full citation, country of origin, study design, summary of key main findings, use made of document in the synthesis and relationship of documents to each other (for example, there may be more than one document reporting on an intervention).

Author response: We have added a summary table as suggested, and we would recommend that this appears as supplementary material.

VERSION 2 – REVIEW

REVIEWER	Helen Cheyne University of Stirling
REVIEW RETURNED	12-Nov-2016

GENERAL COMMENTS	I appreciate that the authors have made efforts to address my initial questions and on reflection and in the light of their revisions I agree with them that the presentation of programme theory now fits in the findings section. I have no further comments.
---

REVIEWER	Rebecca Hardwick University of Exeter Medical School
REVIEW RETURNED	16-Nov-2016

GENERAL COMMENTS	Thank you for the opportunity to review the redrafted paper. I can see the authors have taken into account the feedback from myself and the other peer reviewer. I have a few more points to raise for clarity which I think the authors could address to take the paper from good to excellent. Background It would be great to give an introductory sentence, right at the start, to explain the overarching purpose of HS, and why it is needed, or what the problem is. E.g. women on low incomes have been shown to have less than satisfactory dietary intake, and different interventions have been developed to improve their diets. One of these is Health Start, which has been running in the United Kingdom for the last 10 years. I think this would set up the need for the paper more clearly. At the moment it assumes that the reader already knows the purpose of HS. There is a little confusion now the research questions have been reworded between the stated aim of the review (lines 48-51) and then the research questions. I realise the questions have been reworded to reflect a more 'realist' approach, but I wonder if they still directly relate to the main aim and objective of the review as originally stated? It's important as well to check this throughout the paper to ensure consistency. Start of paragraph 3, on p4, line 55, it would strengthen the rationale
--

to outline why it is that Healthy Start is appropriate to review using realist methods. The authors rightly state that a realist approach was adopted because different individuals respond in different ways to interventions, but thus far it has not established that this is the case for HS. Citing 'conflicting' or 'inconclusive' evidence of the effectiveness of healthy start at this point, and then going on to assert that RR is appropriate because of this would improve the case for using RR.

The authors have done a good job of getting more of the 'meat' of RR as a method and approach into the Background section, on p5. However, I feel that the new sentences read more like statements, rather than a linked explanation of realist approaches. A re-read and potential revise, to make the paragraphs flow more easily would be good. Similarly, I am a bit sad that paragraph starting "The underlying principle is that outcomes (O) are caused etc" has been deleted, as I think this provides a good explanation. In my previous comment about unpacking what is meant by context, I didn't think the authors would delete that paragraph! Is there a third way, of giving a good explanation of how CMOs 'work', and then explaining about context?

Methods/Results

Having had time to reflect on the methods/results section, I hope I can offer some helpful thoughts here, as I found the redrafted paper was almost there but not quite.

I do think the methods are written up clearly but I wonder whether the authors had considered whether it would make sense to, in the methods, describe the process of identifying and building candidate programme theories (currently in results section), and then state the findings of that process and what they are, followed by describing the search, inclusion criteria, study selection, data extraction, quality appraisal, analysis and synthesis methods? The point of the review is to look at the literature as relates to the candidate theories already developed – these candidate theories were not only developed from the literature, which is totally fine, but it is confusing to me not have them spelt out. The other reviewer highlighted this in her review too, and I do not think the authors have adequately addressed this – the candidate programme theories are not stated, either in the methods or results section.

Have a look at how it's set out in this paper under methods – it's essentially a narrative that provides clear information on what they did to develop programme theory, how they did it and why they did it, and finally, what the candidate programme theory was (framework in this instance) :

<http://bmcpublichealth.biomedcentral.com/articles/10.1186/1471-2458-11-222>

Then the results section can focus on the review of the literature AKA the testing of the candidate programme theory, and could run – search findings, and relevance to candidate programme theories.

Results

P 14, lines 16-20. Is there some supporting evidence to the statement that "women's values may change over time etc."? It would be interesting to know what influences the fluctuation between ways of prioritising.

P14 Quote no 1., lines 36-42. Great to use quotes – however, the last line of the quote is confusing, and I am not sure I get the illustration. I am also not sure that quote illustrates how HS Vouchers get used by women who prioritise healthy eating – is there a clearer quote instead? The second quote from the midwife further down the page is excellent to illustrate the point. And excellent quote to illustrate bending the rules too.

	P15 lines 54-58 – potential for slight confusion here – the vouchers are address to them (the women), but they don't have the name of the beneficiary printed on them – is that a contradiction? Does it mean that the vouchers are posted to them at their address, but no name is printed on the voucher? Would be great for it to be reworded and made clearer. Discussion Statement of principle findings – I don't think the first sentence adds anything to the paper – the purpose of the paper was to begin to explain these things, which it has done, and that introductory sentence, for me, is unhelpfully stating the obvious. We already know this – what we want to know is what did the review find out that gets us further than that. Limitations – is it necessary to say that 'only theories that were adequately supported by the existing evidence were presented' here? I don't think that is a limitation of the review, it is just another way of talking about the focus of the review. Comparisons with existing literature and theories – this part of the paper is to cement the findings within the wider literature – to support or modify knowledge of a topic in a field. I don't think it achieves this yet. I think that the review has interesting things to say about resource prioritisation, rule bending and disempowerment, but at the moment, I don't think the section really draws out how these findings relate to the wider literature on those things. I think 'resource prioritisation' has been focussed on in this section, but what about rule bending and disempowerment? I realise word count/space is an issue, but I think with some editing of the two paragraphs, space could be given to the other mechanisms. I think the discussion of incentives is potentially confusing here – the review defined the unintended consequence as financial assistance, not incentive, and I am not sure it adds much to discuss incentives and incentive theory here. Conclusion/Abstract conclusion – second to last sentence says 'further evidence is needed' – hasn't this review answered that question by saying it is women who value healthy eating that will experience the intended outcome of dietary improvements? I don't know if it's necessary to add this sentence. And the final sentence that starts 'targeted support' needs more detail – what has this review shown us or helped us learn about the nature of such targeted support? What kind of support would that be? Who should get it? Why? P25 – the font is different in the lower three boxes of "influence of other family members/disempowerment/vouchers handed over to others" Great to see the Study characteristics table. P37 – keyword – if you have space, also include 'synthesis'
--	--

VERSION 2 – AUTHOR RESPONSE

Author's Response to Decision Letter for (bmjopen-2016-013731.R1)

A realist review to explore how low-income pregnant women use food vouchers from the UK's Healthy Start programme

Reviewer comment: Thank you for the opportunity to review the redrafted paper. I can see the authors have taken into account the feedback from myself and the other peer reviewer. I have a

few more points to raise for clarity which I think the authors could address to take the paper from good to excellent.

Author response: Thank you for taking the time to provide such a thorough second review.

Background

Reviewer comment: It would be great to give an introductory sentence, right at the start, to explain the overarching purpose of HS, and why it is needed, or what the problem is. E.g. women on low incomes have been shown to have less than satisfactory dietary intake, and different interventions have been developed to improve their diets. One of these is Health Start, which has been running in the United Kingdom for the last 10 years. I think this would set up the need for the paper more clearly. At the moment it assumes that the reader already knows the purpose of HS.

Author response: Good idea. We have added two sentences to explain when and why Healthy Start was introduced i.e. to help address income-related health inequalities.

Reviewer comment: There is a little confusion now the research questions have been reworded between the stated aim of the review (lines 48-51) and then the research questions. I realise the questions have been reworded to reflect a more 'realist' approach, but I wonder if they still directly relate to the main aim and objective of the review as originally stated? It's important as well to check this throughout the paper to ensure consistency.

Author response: The wording of the review questions has been adjusted slightly to be consistent with the stated aims in the third paragraph.

Reviewer comment: Start of paragraph 3, on p4, line 55, it would strengthen the rationale to outline why it is that Healthy Start is appropriate to review using realist methods. The authors rightly state that a realist approach was adopted because different individuals respond in different ways to interventions, but thus far it has not established that this is the case for HS. Citing 'conflicting' or 'inconclusive' evidence of the effectiveness of healthy start at this point, and then going on to assert that RR is appropriate because of this would improve the case for using RR.

Author response: We have already stated in the paragraph above that there has been no robust evaluation of the impact of Healthy Start on nutritional outcomes. We have added another sentence describing how previous qualitative research has suggested that women may use vouchers in different ways with different motivations, thereby strengthening our case for using the realist review approach.

Reviewer comment: The authors have done a good job of getting more of the 'meat' of RR as a method and approach into the Background section, on p5. However, I feel that the new sentences read more like statements, rather than a linked explanation of realist approaches. A re-read and potential revise, to make the paragraphs flow more easily would be good. Similarly, I am a bit sad that paragraph starting "The underlying principle is that outcomes (O) are caused etc" has been deleted, as I think this provides a good explanation. In my previous comment about unpacking what is meant by context, I didn't think the authors would delete that paragraph! Is there a third way, of giving a good explanation of how CMOs 'work', and then explaining about context?

Author response: We have reinstated a similar sentence to explain how context, mechanisms and outcomes are linked in the CMO configurations.

Methods/Results

Reviewer comment: Having had time to reflect on the methods/results section, I hope I can offer

some helpful thoughts here, as I found the redrafted paper was almost there but not quite. I do think the methods are written up clearly but I wonder whether the authors had considered whether it would make sense to, in the methods, describe the process of identifying and building candidate programme theories (currently in results section), and then state the findings of that process and what they are, followed by describing the search, inclusion criteria, study selection, data extraction, quality appraisal, analysis and synthesis methods? The point of the review is to look at the literature as relates to the candidate theories already developed – these candidate theories were not only developed from the literature, which is totally fine, but it is confusing to me not have them spelt out. The other reviewer highlighted this in her review too, and I do not think the authors have adequately addressed this – the candidate programme theories are not stated, either in the methods or results section.

Have a look at how it's set out in this paper under methods – it's essentially a narrative that provides clear information on what they did to develop programme theory, how they did it and why they did it, and finally, what the candidate programme theory was (framework in this instance) : <http://bmcpublichealth.biomedcentral.com/articles/10.1186/1471-2458-11-222> Then the results section can focus on the review of the literature AKA the testing of the candidate programme theory, and could run – search findings, and relevance to candidate programme theories.

Author response: This makes sense and we have moved the information about our candidate theories from the Results to the Methods as suggested. We have also added a sentence eluding to the main outcome strands, but we feel the evidence-based explanations belong in the Results.

Note to Editor: We could add more details about candidate CMOs in this section (if you think it is necessary) but we feel it would be repetitive of what follows in the Results section.

Results

Reviewer comment: P 14, lines 16-20. Is there some supporting evidence to the statement that "women's values may change over time etc."? It would be interesting to know what influences the fluctuation between ways of prioritising.

Author response: This statement is not directly evidence-based and we have added the words "we propose that" to clarify this. However, the quotes underneath do give some sense of fluctuation, such as being in a dilemma about whether or not to eat healthy foods.

Reviewer comment: P14 Quote no 1., lines 36-42. Great to use quotes – however, the last line of the quote is confusing, and I am not sure I get the illustration. I am also not sure that quote illustrates how HS Vouchers get used by women who prioritise healthy eating – is there a clearer quote instead? The second quote from the midwife further down the page is excellent to illustrate the point. And excellent quote to illustrate bending the rules too.

Author response: The number of illustrative quotes in the literature was limited. This quote provides evidence for part of the CMO (i.e. vouchers enabled her to afford more vegetables than she would have otherwise bought) and the additional mechanism of prioritisation helps to explain why she did not use the voucher to save money as in the second CMO. We were clear in the Methods that: "Where individual extracts of data only supported part of the CMO configuration, it was necessary to make logical inferences about the complete causal pathways and explanations."

Reviewer comment: P15 lines 54-58 – potential for slight confusion here – the vouchers are address to them (the women), but they don't have the name of the beneficiary printed on them – is that a contradiction? Does it mean that the vouchers are posted to them at their address, but no name is printed on the voucher? Would be great for it to be reworded and made clearer.

Author response: The wording has been improved and clarified in this paragraph.

Discussion

Reviewer comment: Statement of principle findings – I don't think the first sentence adds anything to the paper – the purpose of the paper was to begin to explain these things, which it has done, and that introductory sentence, for me, is unhelpfully stating the obvious. We already know this – what we want to know is what did the review find out that gets us further than that.

Author response: We agree this paragraph should focus on the review findings. The first sentence has been removed.

Limitations – is it necessary to say that 'only theories that were adequately supported by the existing evidence were presented' here? I don't think that is a limitation of the review, it is just another way of talking about the focus of the review.

Author response: This sentence has also been removed.

Reviewer comment: Comparisons with existing literature and theories – this part of the paper is to cement the findings within the wider literature – to support or modify knowledge of a topic in a field. I don't think it achieves this yet. I think that the review has interesting things to say about resource prioritisation, rule bending and disempowerment, but at the moment, I don't think the section really draws out how these findings relate to the wider literature on those things. I think 'resource prioritisation' has been focussed on in this section, but what about rule bending and disempowerment? I realise word count/space is an issue, but I think with some editing of the two paragraphs, space could be given to the other mechanisms. I think the discussion of incentives is potentially confusing here – the review defined the unintended consequence as financial assistance, not incentive, and I am not sure it adds much to discuss incentives and incentive theory here.

Author response: Thank you for this advice. We have used this section to situate our findings in the wider literature on the role of individual agency in public health interventions, and the importance of realist methodology to understand agency (or mechanisms). We have also removed some of the superfluous detail about incentive theory.

Reviewer comment: Conclusion/Abstract conclusion – second to last sentence says 'further evidence is needed' – hasn't this review answered that question by saying it is women who value healthy eating that will experience the intended outcome of dietary improvements? I don't know if it's necessary to add this sentence. And the final sentence that starts 'targeted support' needs more detail – what has this review shown us or helped us learn about the nature of such targeted support? What kind of support would that be? Who should get it? Why?

Author response: The sentence referring to further evidence has been removed – we agree it was unnecessary and detracted from the review findings. We do not know enough from this review to make detailed recommendations about additional support for low-income pregnant women. We have altered the last sentence to reflect further research needs.

Reviewer comment: P25 – the font is different in the lower three boxes of "influence of other family members/disempowerment/vouchers handed over to others"

Author response: We assume this comment refers to Figure 1. The font was the same in the versions we submitted, so we are not sure why it did not appear so in the version you received.

Reviewer comment: Great to see the Study characteristics table.

Author response: No response needed.

Reviewer comment: P37 – keyword – if you have space, also include 'synthesis'

Author response: We are unable to add more than six key words.

VERSION 3 – REVIEW

REVIEWER	Rebecca Hardwick University of Exeter Medical School, United Kingdom
REVIEW RETURNED	23-Nov-2016

GENERAL COMMENTS	Thank you for the opportunity to offer further review of the manuscript. The authors have worked hard and have thoughtfully taken into account previous comments and feedback. It reads really well. There are just two things now which I think need addressing more fully before I would recommend it be accepted for publication. Firstly, in my previous peer review, I invited the authors to consider moving the section about the development of candidate programme theories to the methods section, which they have done, and it reads much better this way I think. I also asked them to state the candidate programme theories at the end of that section, or at least somewhere in the methods, but, unless I am missing something, this hasn't been done. I think it would help to include just a sentence, that tells the reader that the candidate theories concerned "prioritisation of resources, bending the rules of the programme and empowerment and personal agency". The second point is that I still don't feel that the discussion section on Comparisons with existing literature and theories sufficiently draws the parallels between the findings of the review and the wider literature. I think it really needs spelling out that 'our review found this, and that resonates/contradicts previous research which said this...', or words to that effect. I am left finishing the paper not really grasping how the findings have extended our knowledge. Which is a real pity, because the review has found useful and interesting new knowledge. I feel it is still muddled in terms of what this section is driving at – my point in the previous peer review hasn't been addressed, that "I think the discussion of incentives is potentially confusing here – the review defined the unintended consequence as financial assistance, not incentive, and I am not sure it adds much to discuss incentives and incentive theory here." – if the authors are saying that HS doesn't fit the 'incentive' criteria (line 33), then that is to do with definitions, and surely fits in the background section where you're defining the parameters of the review/focus, and if "incentives" are unrelated to HS, then why refer to it at all in the discussion? I think it's a distraction to refer to it here, esp because it wasn't explicitly looked at in the review. If I have missed something here, please let me know. Could the authors consider looking at this section by discussing the following: The review found that the prioritisation of resources was important – so what does the literature say about that in relation to
---

	decision making, or healthy eating? The review found that retailers bent the rules, which implies, potentially, something about how important discretion and rule breaking/bending are to successful PH programme implementation – what does a broader literature say about that? Finally, agency is discussed, which is good, but I think it needs to be explicit how this relates to what was found out about how disempowerment works in HS - at the moment, there is discussion to how agency is important in people responding positively to PH interventions, but did the review find that people with low agency (which isn't defined) are more able to use HS? I don't think it did – I think the review found that although women might not need a lot of agency to receive the vouchers, for some women, they actually needed a lot of agency to use the vouchers as intended because otherwise their relatives would use them! There is opportunity here to draw out, compare and contrast what the review has found compared to previous knowledge about the topic of PH intervention more broadly. I hope these comments are useful and constructive. The paper is really very good but these points need fully addressing so that the whole thing is excellent.
--	--

VERSION 3 – AUTHOR RESPONSE

Reviewer Name: Rebecca Hardwick

Institution and Country: University of Exeter Medical School, United Kingdom

Please state any competing interests or state 'None declared': None declared

Reviewer comment: Thank you for the opportunity to offer further review of the manuscript. The authors have worked hard and have thoughtfully taken into account previous comments and feedback. It reads really well. There are just two things now which I think need addressing more fully before I would recommend it be accepted for publication.

Author response: No response required.

Reviewer comment: Firstly, in my previous peer review, I invited the authors to consider moving the section about the development of candidate programme theories to the methods section, which they have done, and it reads much better this way I think. I also asked them to state the candidate programme theories at the end of that section, or at least somewhere in the methods, but, unless I am missing something, this hasn't been done. I think it would help to include just a sentence, that tells the reader that the candidate theories concerned "prioritisation of resources, bending the rules of the programme and empowerment and personal agency".

Author response: We have added a brief statement about the candidate theories that arose from the development stage, which we then set out to test.

Reviewer comment: The second point is that I still don't feel that the discussion section on Comparisons with existing literature and theories sufficiently draws the parallels between the findings of the review and the wider literature. I think it really needs spelling out that 'our review found this, and that resonates/contradicts previous research which said this...', or words to that effect. I am left finishing the paper not really grasping how the findings have extended our knowledge. Which is a real pity, because the review has found useful and interesting new knowledge.

Author response: Thank you for your comment. On reflection we agree that we have not been clear enough about how our study builds on the existing literature and theories. We have added a paragraph at the beginning of this section to highlight what our review adds to previous studies of Healthy Start. This leads into a more in depth discussion of how each of our programme theories compares to existing literature and theories. We hope that we have now addressed your concern.

Reviewer comment: I feel it is still muddled in terms of what this section is driving at – my point in the previous peer review hasn't been addressed, that "I think the discussion of incentives is potentially confusing here – the review defined the unintended consequence as financial assistance, not incentive, and I am not sure it adds much to discuss incentives and incentive theory here." – if the authors are saying that HS doesn't fit the 'incentive' criteria (line 33), then that is to do with definitions, and surely fits in the background section where you're defining the parameters of the review/focus, and if "incentives" are unrelated to HS, then why refer to it at all in the discussion? I think it's a distraction to refer to it here, esp because it wasn't explicitly looked at in the review. If I have missed something here, please let me know.

Author response: Again, thank you for your suggestion. We have removed most of this discussion about incentives and refocused this section on the economics of decision making – values, prioritisation etc. We have left one sentence explaining how Healthy Start differs from the definition of an 'incentive' because we feel this is relevant to the original assumptions (stated aims) about how the programme would work.

Reviewer comment: Could the authors consider looking at this section by discussing the following: The review found that the prioritisation of resources was important – so what does the literature say about that in relation to decision making, or healthy eating? The review found that retailers bent the rules, which implies, potentially, something about how important discretion and rule breaking/bending are to successful PH programme implementation – what does a broader literature say about that?

Author response: We have developed this section by comparing each of our programme theories with relevant literature: how low-income women make decisions and prioritise resources; reasons why other stakeholders may bend the rules; how disempowerment relates to decision making.

Reviewer comment: Finally, agency is discussed, which is good, but I think it needs to be explicit how this relates to what was found out about how disempowerment works in HS - at the moment, there is discussion to how agency is important in people responding positively to PH interventions, but did the review find that people with low agency (which isn't defined) are more able to use HS? I don't think it did – I think the review found that although women might not need a lot of agency to receive the vouchers, for some women, they actually needed a lot of agency to use the vouchers as intended because otherwise their relatives would use them! There is opportunity here to draw out, compare and contrast what the review has found compared to previous knowledge about the topic of PH intervention more broadly.

Author response: We have clarified that our theory on disempowerment relates to how women's decisions and choices may be constrained by other family members. This leads into the wider discussion about agency in public health interventions (with Healthy Start situated as a mid-agency intervention) and we have suggested ways in which the level of agency required might be reduced in the programme. We have also underscored the role of realist methodology in understanding agency (mechanisms) and the contribution of our review to wider debates about public health interventions.

Reviewer comment: I hope these comments are useful and constructive. The paper is really very good but these points need fully addressing so that the whole thing is excellent.

Author response: Thank you Becky. We feel that your suggestions have strengthened our paper and are grateful for your advice.

VERSION 4 – REVIEW

REVIEWER	Rebecca Hardwick University of Exeter
REVIEW RETURNED	16-Dec-2016

GENERAL COMMENTS	Brilliant work, this is excellent. Well done on producing a clear, thoughtful and informative write up of an important research project.
--